# Electron-Derived Molecular Representation Learning for Real-World Molecular Physics

**Gyoung S. Na**[1,†], **Chanyoung Park**[2,†]
[1]Korea Research Institute of Chemical Technology
[2]Korea Advanced Institute of Science and Technology
ngs0@krict.re.kr, cy.park@kaist.ac.kr

## Abstract

Various representation learning methods for molecular structures have been devised to accelerate data-driven drug and materials discovery. However, the representation capabilities of existing methods are essentially limited to *atom*-level information, which is not sufficient to describe real-world molecular physics. Although *electron*-level information can provide fundamental knowledge about chemical compounds beyond the atom-level information, obtaining the electron-level information in real-world molecules is computationally impractical and sometimes infeasible. We propose a new method for learning electron-derived molecular representations without additional computation costs by transferring pre-calculated electron-level information about small molecules to large molecules of our interest. The proposed method achieved state-of-the-art prediction accuracy on extensive benchmark datasets containing experimentally observed molecular physics.

## 1 Introduction

Graph neural networks (GNNs) [1] have been successfully applied to predict the physical and chemical properties of molecules based on graph representations of molecular structures. Typically, a molecular structure is represented as an attributed graph $G = (\mathcal{V}, \mathcal{U}, \mathbf{X}, \mathbf{E})$, where $\mathcal{V}$ is a set of nodes (i.e., atoms), $\mathcal{U}$ is a set of edges (i.e., chemical bonds), $\mathbf{X} \in \mathbb{R}^{|\mathcal{V}| \times d}$ is a $d$-dimensional node-feature matrix, and $\mathbf{E} \in \mathbb{R}^{|\mathcal{U}| \times l}$ is an $l$-dimensional edge-feature matrix [2]. Based on the graph representation, GNNs were able to learn latent molecular embeddings that encode the local geometry of the atoms as well as the physical characteristics of the entire molecular structure [3, 4].

Various GNN-based methods have been proposed to learn informative molecular representation from various approaches, such as molecular geometry [5], fragment-based learning [6, 7], and domain knowledge integration [8]. However, their representation capabilities are fundamentally limited to the atom-level molecular structures, and they overlooked the principle of quantum mechanics that the physical and chemical characteristics of molecules fundamentally originate from the electron structures [9, 10]. Learning molecular representations from the electron-level structure is crucial because many physical and chemical characteristics of molecules are, in fact, essentially derived from their electronic structures [10]. Therefore, we argue that the GNN-based methods would benefit from considering the molecules in the *electron-level* structure beyond the atom-level structure.

A straightforward and direct solution would be to directly provide the electron-level information about the molecules to GNNs by calculating the electronic structures of the molecules based on the calculation methods in computational physics and chemistry, such as density functional theory (DFT) [10]. However, this solution is impractical and sometimes infeasible in real-world large molecules because the calculation methods suffer from cubic or greater time complexities [10] and

---

[†]These authors contributed equally to this work

37th Conference on Neural Information Processing Systems (NeurIPS 2023).

local convergences in the electronic structure calculations [11]. Therefore, we need a new approach for learning electron-derived representations of large molecules without additional calculations and experiments of quantum mechanics.

We propose *hierarchical electron-derived molecular learning* (HEDMoL) to learn molecular representations from the input atom-level molecular structures under their estimated electronic structures. The main idea is to estimate the electron-level information about a large input molecule, which is given in an atom-level structure, by extending the electron-level information about small molecules that compose the large input molecule. As the electron-level information about small molecules is already computed and provided in public calculation databases, we can relieve the computational burden of expensive electronic structure calculations required for large molecules. More precisely, HEDMoL learns electron-derived molecular representations from both the input atom-level molecular information and the estimated electron-level information through the following three steps: (1) HEDMoL decomposes an input atom-level molecular structure into several atom-level substructures based on graph decomposition algorithms. (2) HEDMoL constructs an electron-derived molecular graph by transferring electron-level attributes stored in an external calculation database to each of the decomposed substructures based on structural similarities between the decomposed substructures and the small molecules in the external calculation database, and we call this process *knowledge extension*. (3) HEDMoL generates molecular representations through a hierarchical representation learning on the latent atom- and electron-level information.

It is worth noting that in this study, we focus on evaluating the prediction capabilities of the prediction models on *experimental* datasets rather than *calculation* datasets (e.g., QM9 dataset [12]). Although the calculation datasets are useful for analyzing rough relationships between atomic geometry and molecular properties, the calculation datasets are not appropriate to evaluate the prediction capabilities of machine learning methods on real-world molecular physics because they do not sufficiently simulate the uncertainty and complex configurations in quantum mechanics [13, 14]. For these reasons, we collected eight experimentally-generated molecular datasets from physicochemistry, toxicity, and pharmacokinetics applications. In the experiments, we measured the prediction accuracy of the prediction models on the experimental molecular datasets to evaluate the prediction capabilities of the prediction models on complex *real-world molecular physics*. HEDMoL achieves state-of-the-art performance in predicting the experimentally observed physical and chemical properties of the molecules. Furthermore, HEDMoL outperforms state-of-the-art GNNs in various regression tasks on small training datasets, which is one of the main challenges of machine learning in chemical applications [15, 16].

## 2 Method

### 2.1 Problem Reformulation on Electronic Substructures

First of all, we reformulate the prediction problem on molecular structures as a problem on decomposed molecular substructures. By doing so, we can extend the knowledge of small molecules, which is not expensive to calculate or measure the electronic attributes, into the knowledge of complex real-world molecules. This knowledge extension from theoretically calculated information about small atomic structures to real-world experimental observations is a long-standing challenge in computational physics and chemistry [17, 18]. Physically, we can define a problem to predict the molecular properties $y$ as follows.

$$y = (f \circ g)(\mathcal{E}), \tag{1}$$

where $\mathcal{E}$ is the electronic structure of a molecule, $g$ is a physics-informed function to generate a numerical representation from $\mathcal{E}$, and $f$ is a function to calculate the physical and chemical properties from the electron-level descriptors $g(\mathcal{E})$. Since the calculation methods to calculate the electronic structure $\mathcal{E}$ has a cubic or greater time complexity with respect to the number of atoms, existing GNNs-based methods basically assume that $g(\mathcal{E})$ can be sufficiently approximated by the atom-level molecular structures to avoid the impractical time complexity in the electronic structure calculation. However, the physical information in the electronic structures is inevitably distorted and simplified in the process of converting $g(\mathcal{E})$ to the atom-level molecular structures [10, 19].

Instead of approximating $\mathcal{E}$ with the atom-level structures, we reformulate the problem in Eq. (1) as a problem on a set of small substructures by decomposing the input electronic structures $\mathcal{E}$ into $K$

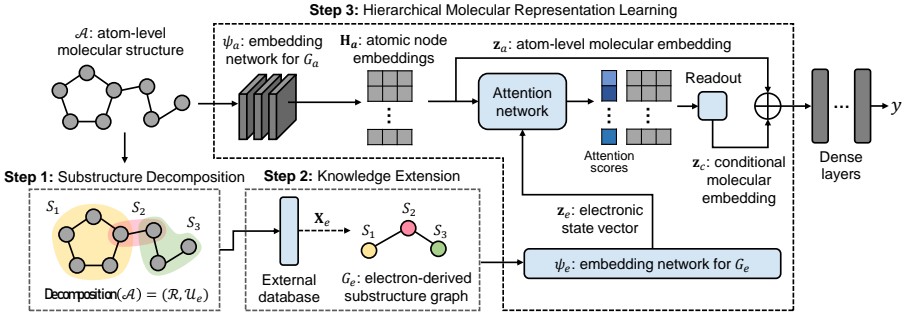

Figure 1: The overall process of HEDMoL to predict the target molecular property $y$ of the input atom-level molecular structure $\mathcal{A}$. $\mathcal{R} = \{S_1, S_2, S_3\}$: a set of the decomposed atom-level substructures. $\mathcal{U}_e$: a set of edges between the decomposed atom-level substructures. $\mathbf{z}_a$ and $\mathbf{z}_c$: calculated molecular embeddings of $\mathcal{A}$, which are defined in Eq. (8)

substructures as:

$$y = (f \circ g_G)(\{g_L(\mathcal{E}_1), ..., g_L(\mathcal{E}_K)\}), \tag{2}$$

where $\mathcal{E}_k \in \mathcal{T}(\mathcal{E})$ is the $k$-th electronic substructure of $\mathcal{E}$, $\mathcal{T}(\mathcal{E})$ in a set of physically possible electronic substructures of $\mathcal{E}$, $g_L$ is a descriptor function for the electronic substructures $\mathcal{E}_k$, and $g_G$ is an order-invariant function to generate the molecular representation from a set of electron-level descriptors $\{g_L(\mathcal{E}_1), ..., g_L(\mathcal{E}_K)\}$. HEDMoL aims to predict the target physical and chemical properties of molecules based on the decomposed formulation in Eq. (2) without the expensive electronic structure calculations on the entire molecule.

## 2.2 Overall Architecture of HEDMoL

Fig. 1 illustrates the overall prediction process of HEDMoL, which consists of three steps as: (1) Substructure decomposition, (2) Knowledge extension, and (3) Hierarchical molecular representation learning. Each step of HEDMoL is summarized as follows. **(Step 1)** An input atom-level molecular structure $\mathcal{A}$ is decomposed into a set of atom-level substructures $\mathcal{R} = \{S_1, ..., S_K\}$. Each decomposed substructure $S_k$ implies the atom-level representation of the decomposed electronic substructures $\mathcal{E}_k$ and will be converted into $g_L(\mathcal{E}_k)$ in the next knowledge extension step. **(Step 2)** HEDMoL assigns electron-level attributes in external calculation databases to each decomposed substructure $S_k$, i.e., the decomposed atom-level substructures $\mathcal{R}$ is converted into a set of electron-level descriptors $\{g_L(\mathcal{E}_1), ..., g_L(\mathcal{E}_K)\}$ by transferring pre-calculated electronic attributes in an external calculation database. **(Step 3)** HEDMoL calculates an electronic state vector by $\mathbf{z}_e = \psi_e(G_e)$, where $\psi_e$ is a GNN-based embedding network, and $G_e$ is a graph representation of $\{g_L(\mathcal{E}_1), ..., g_L(\mathcal{E}_K)\}$. For the electronic state vector $\mathbf{z}_e$, HEDMoL generates two molecular embeddings from atomic and electron-conditioned structures to predict the target molecular property $y$. In the following sections, we will explain the implementation details of each step in HEDMoL.

## 2.3 Substructure Decomposition

The substructure decomposition of HEDMoL is a data pre-processing step to convert an input atom-level molecular structure into a tuple $(\mathcal{R}, \mathcal{U}_e)$, where $\mathcal{R} = \{S_1, ..., S_K\}$ is a set of the decomposed substructures of an input atom-level molecular structure $\mathcal{A}$, $K$ is the number of decomposed substructures, $\mathcal{U}_e$ is a set of edges between the decomposed substructures. Physically, each decomposed substructure $S_k$ implies the atom-level representation of the decomposed electronic substructures $\mathcal{E}_k$ in Eq. (2). In the decomposition process, we enforce a constraint $\mathcal{A} = S_1 \cup ... \cup S_K$ to prevent information loss from the substructure decomposition. In the next knowledge extension step, the decomposed substructures will be converted into a set of electron-level descriptors $\{g_L(\mathcal{E}_1), ..., g_L(\mathcal{E}_K)\}$.

Among various choices for the implementation of the substructure decomposition of HEDMoL, such as spectral clustering [20], BRICS decomposition [21], and junction tree algorithm [22], we employed the junction tree algorithm due to the following three benefits of this algorithm as: (1) The junction tree algorithm does not require a hyperparameter tuning for each input graph. (2) It provides generalized graph abstraction and clustering results of the molecular structures [22]. (3) It satisfies our

constraint $\mathcal{A} = S_1 \cup ... \cup S_K$ in the substructure decomposition. Therefore, HEDMoL decomposes the input atom-level molecular structure $\mathcal{A}$ into a set of atom-level substructures $\mathcal{R} = \{S_1, ..., S_K\}$ based on the junction tree algorithm, where $S_k$ is a vertex clique in the entire junction tree of $\mathcal{A}$. The experimental comparison of HEDMoL with the junction tree and BRICS decomposition algorithms are provided in Appendix 5 of Supplementary Material.

## 2.4 Knowledge Extension

To avoid the computational bottleneck in the electronic structure calculation, the knowledge extension step of HEDMoL aims to transfer the electron-level information from an external calculation database $\mathcal{D}_s$ to each of the decomposed substructures $S_k \in \mathcal{R}$ instead of calculating the electronic structures. That is, a set of the atom-level decomposed substructures $\mathcal{R}$ is converted into a set of electron-level descriptors $\{g_L(\mathcal{E}_1), ..., g_L(\mathcal{E}_K)\}$. By our reformulated problem in Eq. (2), HEDMoL can utilize the knowledge about the small molecules from the external calculation databases and extend the knowledge to large molecules without additional calculations and architectural modifications.

In the knowledge extension step, HEDMoL matches the decomposed substructures to the small molecules in an external calculation database by calculating the molecular distance between them. Based on the calculated molecular distances, HEDMoL transfers the knowledge of the external calculation database to each decomposed substructure as:

$$\mathbf{X}_{e,k} = \mathbf{q}_{\mathtt{idx}_k}, \tag{3}$$

where $\mathbf{X}_{e,k} \in \mathbb{R}^q$ is the $k$-th row vector of $\mathbf{X}_e \in \mathbb{R}^{K \times q}$, and it denotes the node feature of the $k$-th decomposed substructure $S_k$. $\mathbf{q}_{\mathtt{idx}_k} \in \mathbb{R}^q$ is the pre-calculated electron-level attributes of a small molecule in $\mathcal{D}_s$, where $\mathtt{idx}_k$ is the index (such as) of the small molecule in $\mathcal{D}_s$ that is the most similar to $S_k$. Formally, $\mathtt{idx}_k$ is calculated by:

$$\mathtt{idx}_k = \underset{i \in \{1, 2, ..., |\mathcal{D}_s|\}}{\arg\min} \pi(S_k, G_i^s), \tag{4}$$

where $G_i^s$ is the atom-level graph of the $i$-th molecule in $\mathcal{D}_s$, $\pi$ is a distance metric between two graphs. Among several choices, we implemented $\pi$ based on a robust and efficient unsupervised graph embedding method called geometric scattering on graphs (GeoScattering) [23], and the molecular distance was calculated by the Euclidean distance between the GeoScattering embeddings of $S_k$ and $G_i^s$. The prediction capabilities of HEDMoL for the implementations of $\pi$ with different graph embedding methods are experimentally evaluated in Appendix 6 of Supplementary Material.

After the knowledge extension step, an electron-derived substructure graph $G_e$ of the input molecule is generated by reconstructing an attributed graph based on the decomposed substructures and the assigned electron-level attributes as $G_e = (\mathcal{R}, \mathcal{U}_e, \mathbf{X}_e)$. Thus, we generated $G_e$ containing fragmented information about the electronic structure without expensive electronic structure calculations. In the next step, HEDMoL generates a latent molecular embedding through a hierarchical representation learning of the atom-level molecular graph $G_a = (\mathcal{V}, \mathcal{U}, \mathbf{X}_a, \mathbf{E})$ and the electron-derived substructure graph $G_e$ of the input molecule, where $\mathbf{X}_a$ is an input atom-feature matrix.

## 2.5 Hierarchical Molecular Representation Learning

The purpose of the hierarchical representation learning in HEDMoL is to generate a latent embedding vector $\mathbf{z}$ of the input molecule by propagating the electron-level information from a latent embedding of $G_e$ to the embedding process of the atom-level molecular descriptor $G_a$. This mechanism is consistent with the physical principle because the atomic configurations fundamentally depend on the electronic structures [9, 10]. In the hierarchical representation learning, we employed a GNN-based embedding network $\psi_e$ for an order-invariant representation learning on a set $\{g_L(\mathcal{E}_1), ..., g_L(\mathcal{E}_K)\}$. Formally, HEDMoL calculates an order-invariant electron-derived embedding $\mathbf{H}_e$ on $\{g_L(\mathcal{E}_1), ..., g_L(\mathcal{E}_K)\}$ as:

$$\mathbf{H}_e = \psi_e(G_e) \approx g_G(\{g_L(\mathcal{E}_1), ..., g_L(\mathcal{E}_K)\}). \tag{5}$$

For the calculated $\mathbf{H}_e$, an electronic state vector $\mathbf{z}_e$ is calculated by $\mathbf{z}_e = \sum_{i=1}^{|\mathcal{R}|} \mathbf{H}_{e,i}$, where $\mathbf{H}_{e,i}$ is the $i$-th row-vector of $\mathbf{H}_e$. After the embedding process of $G_e$, GNN-based embedding network $\psi_a$ calculates an atom-level node embedding matrix as $\mathbf{H}_a = \psi_a(G_a)$ for a given atom-level molecular

graph $G_a$. Then, an atom-level molecular embedding $\mathbf{z}_a$ is similarly calculated by $\mathbf{z}_a = \sum_{i=1}^{|\mathcal{V}|} \mathbf{H}_{a,i}$. On the other hand, a conditional atom-level molecular embedding $\mathbf{z}_c$ is calculated by considering the atomic contributions under the given electronic state vector $\mathbf{z}_e$ as:

$$\mathbf{z}_c = \sum_{i=1}^{|\mathcal{V}|} \frac{\exp(f_a(\mathbf{H}_{a,i} \oplus \mathbf{z}_e))}{\sum_{j=1}^{|\mathcal{V}|} \exp(f_a(\mathbf{H}_{a,j} \oplus \mathbf{z}_e))} \mathbf{H}_{a,i}, \tag{6}$$

where $f_a$ is a trainable neural network to calculate attention scores of the $i$-th atom under the given electronic structure described by the electronic state vector $\mathbf{z}_e$. In other words, $\mathbf{z}_c$ contains atom-level information conditioned by underlying electronic structures.

After the graph embedding process, HEDMoL generates the molecular embedding $\mathbf{z}$ by concatenating the graph-level embeddings from different views as $\mathbf{z} = \mathbf{z}_a \oplus \mathbf{z}_c$, where $\mathbf{z}_a$ is the graph-level embedding of $G_a$. Conceptually, the molecular embedding $\mathbf{z}$ contains the atomic information conditioned by electronic structures as well as the atom-level information about the input molecules. Finally, HEDMoL predicts the target molecular properties $y = f_d(\mathbf{z})$ by entering $\mathbf{z}$ to the trainable dense layers $f_d$.

## 2.6 Energy-Based Physical Consistency Regularization

In the molecular representation learning of HEDMoL, we enforce that the latent atom- and electron-level embeddings of the input molecule to indicate the same potential energy, which is one of the universal quantities to describe the atomic systems [10, 24, 25]. Physically, the atom- and electron-level descriptors of a molecule should have the same potential energy because they describe essentially the same molecular structure. To this end, we introduce two constraints on the input molecule as:

$$E_{p,k} + \epsilon_k = E_{a,k}, \text{s.t.} E_{a,k} = E_{e,k}, \forall k = 1, 2, ..., |\mathcal{R}|, \tag{7}$$

where $E_{p,k}$ is the calculated physical energy of the small molecule that matches with the $k$-th decomposed substructure $S_k$, $\epsilon_k \sim \mathcal{N}(0, \sigma_k)$ is independent and identically normally distributed random variables following a normal distribution $\mathcal{N}(0, \sigma_k)$, $E_{a,k}$ is the predicted energy from the node embeddings of the atoms in $S_k$, and $E_{e,k}$ is the predicted energy from the $k$-th substructure embedding $\mathbf{H}_{e,k}$. $\epsilon_k$ indicates the approximation error in transferring the physical energy of the small molecule in $\mathcal{D}_s$ to the decomposed substructures in $\mathcal{R}$.

Physically, the energy of an atomic system $\mathcal{A}$ can be described by the many-body potential energies of the atoms in $\mathcal{A}$ as [26]:

$$E = \sum_{i_1, i_2}^{|\mathcal{A}|} V_2(\mathcal{H}_{i_1}, \mathcal{H}_{i_2}) + \sum_{i_1, i_2, i_3}^{|\mathcal{A}|} V_3(\mathcal{H}_{i_1}, \mathcal{H}_{i_2}, \mathcal{H}_{i_3})$$
$$+ \cdots + \sum_{i_1, ..., i_{|\mathcal{A}|}}^{|\mathcal{A}|} V_{|\mathcal{A}|}(\mathcal{H}_{i_1}, ..., \mathcal{H}_{i_{|\mathcal{A}|}}), \tag{8}$$

where $|\mathcal{A}|$ is the number of atoms in the atomic system $\mathcal{A}$, $V_i$ is the $i$-body potential function, and $\mathcal{H}_i$ is the local environment around the $i$-th atom. However, calculating the many-body potential energy is infeasible due to the computational complexity. To overcome the infeasible computational complexity, we approximate the many-body potential energy in Eq. (8) based on a trainable message-passing function, as the message-passing scheme is an efficient approach for predicting the physical properties from the physical interactions of particles [27, 28]. We define the many-body potential energy of $S_k$ based on the graph self-attention mechanism [29], which calculates the interatomic attention score $\alpha_{i,j}$. Formally, HEDMoL predicts the many-body potential energy of $S_k$ based on a trainable energy function $f_e$ and a message-passing function $g_e$ as:

$$E_{a,k} = f_e(g_e(S_k)), \tag{9}$$

where a vector-shaped atom-level substructure embedding $g_e(S_k)$ is given by

$$g_e(S_k) = \frac{1}{|S_k|} \sum_{i \in S_k} \left( \mathbf{W}\mathbf{H}_{a,i} + \sum_{j \in \mathcal{N}_i \cap S_k} \alpha_{i,j} \mathbf{V}\mathbf{H}_{a,j} \right), \tag{10}$$

$|S_k|$ is the number of atoms in a substructure $S_k$, $\mathbf{W}$ and $\mathbf{V}$ are trainable weight matrices of $g_e$, $\mathcal{N}_i$ is a set of indices of the atoms connected to the $i$-th atom, and $\alpha_{i,j}$ is an attention score between the $i$-th and $j$-th atoms. Similarly, we define $E_{e,k}$ of the $k$-th decomposed substructure based on the trainable energy function $f_e$ in Eq. (9) as:

$$E_{e,k} = f_e(\mathbf{H}_{e,k}) \tag{11}$$

Based on Eqs. (9) and (11), we define two regularization terms $\Omega_{a,n}$ and $\Omega_{e,n}$ for the $n$-th molecule in the training dataset $\mathcal{D}$ as:

$$\Omega_{a,n} = \sum_{k=1}^{|\mathcal{R}_n|} \max\{|E_{a,k} - f_e(g_e(S_k))| - \alpha, 0\}, \tag{12}$$

$$\Omega_{e,n} = \sum_{k=1}^{|\mathcal{R}_n|} \max\{|E_{e,k} - f_e(\mathbf{H}_{e,k}))| - \alpha, 0\}, \tag{13}$$

where $\mathcal{R}_n$ is the set of decomposed substructures of the $n$-th molecule, $\alpha \geq 0$ is a hyperparameter to allow uncontrollable energy differences incurred by the structural differences between the decomposed substructures and the matched small molecules in the external calculation database. The hyperparameter $\alpha$ is a physically bounded variable [14], and the energy differences between small organic molecules are usually in a range from 0.1 to 0.3 electronvolts because the energy that the small organic molecules can have is physically bounded [9, 14]. Finally, we optimize the model parameters of HEDMoL to minimize the following loss function $L$ as:

$$L = \sum_{n=1}^{|\mathcal{D}|} L_p(y_n, f_d(\mathbf{z}_n)) + \lambda \left( \sum_{n=1}^{|\mathcal{D}|} \Omega_{a,n} + \Omega_{e,n} \right), \tag{14}$$

where $L_p$ is a prediction loss, and $\lambda \geq 0$ is a hyperparameter to control the effect of the regularization terms $\Omega_{a,n}$ and $\Omega_{e,n}$ in the training of HEDMoL. Experimental evaluations of HEDMoL for different values of $\alpha$ and $\lambda$ are provided in Appendix 7 of Supplementary Material.

## 3  Experiments

We compared the prediction accuracies of HEDMoL with those of the state-of-the-art methods on various benchmark molecular datasets containing experimentally observed molecular physics. The experimental molecular datasets were provided by public chemical databases, such as MoleculeNet [30] and ChEMBL [31]. We selected eight benchmark datasets from various application fields including physicochemistry, toxicity, and pharmacokinetics, as shown in Appendix 2 of Supplementary Material. We compared the prediction capabilities of HEDMoL with those of a tree-based ensemble method [32] and ten state-of-the-art GNNs [33–39, 3, 40, 41]. We generated XGB-Mor, XGB-FC, and XGB-MK that predict target molecular properties for input Morgan (Mor) [42], functional-class (FC) [42], and MACCS Key (MK) [43] fingerprints, respectively. Although the 3D structure-based GNNs are not applicable to the experimental molecular datasets due to the absence of the 3D atomic coordinates, we additionally calculated the 3D atomic coordinates based on the semi-empirical method in RDKit[1] and evaluated the 3D structure-based GNNs based on the generated 3D atomic coordinates. However, we were not able to execute or evaluate PaiNN [44], GemNet [45], and Equiformer [4] on the experimental molecular datasets due to the out-of-memory problem or required additional information.

In the experiments, we focused on evaluating the prediction capabilities of HEDMoL on the *experimental* datasets rather than *calculation* datasets, due to the following two reasons: (1) Evaluations on the calculation datasets (e.g., QM9 dataset) are not fair because HEDMoL exploits the external calculation databases in the knowledge extension step. (2) The experimental datasets containing the uncertainty of the atomic systems are closer to real-world molecular physics than the calculation datasets [46]. We used SchNet, MPNN, and GIN as the GNN-based embedding networks of HEDMoL. Implementation details and hyperparameter settings of HEDMoL are provided in Appendix 4 of Supplementary Material. We used a subset of the QM9 dataset containing the molecules of

---

[1]https://www.rdkit.org/

Table 1: Measured $R^2$-scores on the benchmark molecular datasets. Input type means the required data format of the input molecules. The highest $R^2$-score for each benchmark dataset has been remarked in bold, and the standard deviation of the $R^2$-scores is presented in parentheses. N/R means the negative R2-score indicating the failure of the machine learning model. N/A is "not available" that means the out-of-memory problem or impractical computation cost.

| Input Type | Method | Lipop | ESOL | ADMET | IGC50 | LC50 | LD50 | LMC-H | LMC-R |
|---|---|---|---|---|---|---|---|---|---|
| Molecular Fingerprint | XGB-Mor [42] | 0.531 (0.024) | 0.659 (0.045) | 0.717 (0.021) | 0.621 (0.040) | 0.390 (0.133) | 0.497 (0.016) | 0.505 (0.018) | 0.617 (0.058) |
| | XGB-FC [42] | 0.578 (0.018) | 0.686 (0.052) | 0.720 (0.009) | 0.628 (0.023) | 0.501 (0.052) | 0.519 (0.025) | 0.503 (0.007) | 0.612 (0.015) |
| | XGB-MK [43] | 0.542 (0.041) | 0.764 (0.047) | 0.761 (0.020) | 0.680 (0.037) | 0.486 (0.112) | 0.526 (0.021) | 0.471 (0.019) | 0.591 (0.033) |
| 3D molecular Graph | DimeNet++ [33] | N/R | 0.878 (0.025) | N/R | 0.779 (0.019) | 0.591 (0.055) | N/A | 0.352 (0.101) | N/R |
| | PhysChem [34] | 0.694 (0.024) | 0.848 (0.032) | N/A | 0.814 (0.017) | 0.511 (0.053) | N/A | N/A | N/R |
| | M3GNet [35] | N/A | 0.857 (0.025) | N/A | 0.697 (0.029) | 0.531 (0.034) | N/A | N/A | 0.565 (0.041) |
| 2D Molecular Graph | GIN [36] | 0.702 (0.031) | 0.897 (0.022) | 0.833 (0.017) | 0.799 (0.021) | 0.543 (0.080) | 0.515 (0.044) | 0.443 (0.027) | 0.568 (0.020) |
| | EGC [37] | 0.708 (0.043) | 0.896 (0.017) | 0.838 (0.012) | 0.808 (0.029) | 0.575 (0.045) | 0.497 (0.034) | 0.441 (0.023) | 0.566 (0.017) |
| | MPNN [38] | 0.711 (0.022) | 0.894 (0.023) | 0.830 (0.014) | 0.797 (0.018) | 0.532 (0.064) | 0.469 (0.040) | 0.449 (0.057) | 0.564 (0.031) |
| | CGCNN [39] | 0.701 (0.034) | 0.899 (0.029) | 0.836 (0.008) | 0.807 (0.018) | 0.531 (0.040) | 0.482 (0.041) | 0.436 (0.051) | 0.588 (0.020) |
| | SchNet [3] | 0.667 (0.021) | 0.881 (0.026) | 0.834 (0.012) | 0.765 (0.034) | 0.587 (0.052) | 0.467 (0.025) | 0.456 (0.024) | 0.543 (0.033) |
| | MEGNet [40] | 0.604 (0.023) | 0.889 (0.027) | 0.826 (0.024) | 0.754 (0.026) | 0.574 (0.122) | 0.505 (0.027) | 0.422 (0.032) | 0.617 (0.058) |
| | UniMP [41] | 0.702 (0.030) | 0.886 (0.025) | 0.833 (0.014) | 0.793 (0.027) | 0.504 (0.031) | 0.470 (0.025) | 0.422 (0.061) | 0.579 (0.036) |
| | **HEDMoL** | **0.759 (0.043)** | **0.914 (0.016)** | **0.865 (0.014)** | **0.840 (0.010)** | **0.663 (0.053)** | **0.572 (0.035)** | **0.551 (0.008)** | **0.639 (0.035)** |

maximum six atoms as an external calculation database for knowledge extension of HEDMoL, as described in Appendix 4 of Supplementary Material. The source code of HEDMoL and the experiment scripts are publicly available at `https://github.com/anonauthor60/HEDMoL`. The experimental evaluations of HEDMoL for different choices of implementations and external databases are provided in Supplementary Material.

### 3.1   Prediction Accuracy on Experimentally Collected Benchmark Datasets

We measured $R^2$-scores of the state-of-the-art competitors and HEDMoL based on the leave-one-out 5-fold cross-validation so that the test dataset covers all molecules in the original dataset because the prediction accuracy of the prediction models on the real-world chemical data is sensitive to training/test split [47]. We reported the mean of the measured $R^2$-scores with the standard deviation on the test datasets. Table 1 presents the measured $R^2$-scores and standard deviations on the eight benchmark molecular datasets, and HEDMoL achieved the best $R^2$-scores for all benchmark datasets. HEDMoL outperformed the competitor methods over the standard deviations for all benchmark datasets except for the ESOL dataset. In particular, HEDMoL showed higher $R^2$-scores than individual GIN, EGC, and SchNet, which are used as the embedding networks of HEDMoL. Furthermore, HEDMoL outperformed the 3D-based GNNs, DimeNet++, PhysChem, and M3GNet, even though they used additional 3D atomic coordinates. This result stems from the approximation and calculation errors in the 3D molecular generation. Experimental results on MAEs were consistent with the results in Table 1, as shown in Appendix 10. These results show that HEDMoL learned more generalized and informative molecular representations beyond individual GNNs through the hierarchical representation learning on $G_a$ and $G_e$.

In the experiment, GNNs outperformed the XGB-based models in the problems of predicting the physicochemistry properties on the Lipop, ESOL, and ADMET datasets. However, the simple XGB-based models showed higher $R^2$-scores than those of the state-of-the-art GNNs on the LMC-H and LMC-R datasets containing relatively large molecules. This result is consistent with an experimental

observation in a previous study [48] regarding the overfitting problems of GNNs on large atomic systems. As shown in the experimental results, the fingerprint-based models and GNNs have their own limitations when applied to the experimental molecular data. The fingerprint-based methods suffer from the lack of physical information about the input molecules because the molecular fingerprints are designed to describe the connectivities of the atoms rather than representing the physical attributes of each atom. On the other hand, although GNNs can extract physical information from the input molecular graphs containing physical attributes of each atom, they can be easily overfitted in large atomic systems [48]. However, HEDMoL overcomes both limitations of the existing methods by exploiting the molecular graph with electronic attributes, which are robust to extrapolation [9, 10, 17]. The prediction accuracy and improvement of HEDMoL according to the molecular sizes were presented in Appendix 12 of Supplementary Material. The experimental results demonstrate that HEDMoL can provide more accurate and robust prediction results on real-world molecular physics without additional electronic structure calculations.

## 4   Conclusion

This paper proposed HEDMoL for learning electron-derived molecular representations to improve the prediction capabilities on real-world molecular physics. HEDMoL learned the electron-derived molecular representations without additional calculation costs by transferring the pre-calculated electron-level information of small molecules in an external database to large input molecules. HEDMoL achieved state-of-the-art prediction accuracy on extensive molecular datasets containing experimentally observed molecules and their properties. Furthermore, HEDMoL showed better prediction accuracies even under the lack of training data, which is one of the main challenges of machine learning in chemical applications. These results showed the practical potential of HEDMoL in real-world chemical applications.

## Acknowledgement

This research was supported by Korea Evaluation Institute of Industrial Technology (No. TS231-10R) and Korea Research Institute of Chemical Technology (No. KK2351-10). This work was supported by Institute of Information & communications Technology Planning & evaluation (IITP) grant funded by the Korea government (MSIT) (No. 2022-0-00077)

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

## Appendix 1: Related Work

**Graph Neural Networks on Molecular Structures**

Various GNNs were proposed to predict the physical and chemical properties of molecules [2, 49]. SchNet [3] and MEGNet [40] are graph convolutional networks to learn quantum chemical principles in atomic structures based on atom-wise representations of molecules. MPNN introduced in [38] is a message-passing neural network for learning quantum mechanics in atomic structures. CGCNN [39] is a graph convolutional network designed for processing crystalline materials and molecules, and it showed state-of-the-art performance in various chemical applications on molecular datasets. DimeNet++ [33], PhysChem [34], and M3GNet [35] were devised based on message-passing schemes that generate molecular embeddings by propagating the local and global 3D atomic geometry. In addition to these methods, PaiNN [44], GemNet [45], and Equiformer [4] were proposed for molecular property prediction on the 3D structures. However, despite their state-of-the-art performances on several benchmark datasets of calculated molecular structures, their applicability in real-world molecular physics is limited in most cases because it is hard to accurately determine the 3D atomic coordinates in real-world chemical compounds due to the uncertainty of atomic positions and the limitations of measurement equipment [5, 50].

**Transfer Learning on Molecular Datasets**

Machine learning in chemical applications usually suffers from the lack of training data because conducting chemical experiments to collect the training data is expensive and time-consuming [51, 52]. To overcome the lack of the training data, transfer learning has received significant interest in physics and chemistry [15, 16, 53]. Several transfer learning methods were successfully applied to various chemical applications by transferring pre-trained models on large source calculation databases to target experimental datasets [16, 53]. Nonetheless, the prediction capability of existing transfer learning methods is inherently limited because the source calculation databases are not able to cover the majority of large molecules in real-world experimental datasets due to the cubic or greater time complexities of the calculation methods with respect to the number of electrons in the molecules [9, 10].

In addition, various molecular representation learning methods were proposed to learn informative molecular representations by transferring knowledge of decomposed substructures to the entire molecules [8, 6, 54, 55, 7]. However, the representation capabilities of the existing methods are inherently limited to the atom-level information, and they did not consider the fundamental relationships between the molecular properties and the electronic structures of the molecules [8, 6, 54, 55, 7]. Moreover, some existing method requires additional information about substructure-based synthesis recipe of the molecule [8], which is not available in real-world molecules. Therefore, we need a new approach for transferring electron-level information between the molecules of different scales without additional calculations and experiments.

## Appendix 2: Benchmark Molecular Datasets

We evaluated the prediction capabilities of HEDMoL on various benchmark datasets containing experimentally observed molecular properties from chemical databases, such as MoleculeNet [30] and ChEMBL [31]. We selected eight benchmark datasets from various application fields, including physicochemistry, toxicity, and pharmacokinetics, as shown in Table 2.

## Appendix 3: Competitor Methods

In the experiments, we compared the prediction capabilities of HEDMoL with a baseline tree method and eight state-of-the-art GNNs, which have been widely used in chemical applications. The competitor methods are briefly described as:

- **XGB-Mor**: XGBoost (XGB) [32] is a tree-based gradient boosting model, and it showed state-of-the-art performances in various scientific applications. For the experimental evaluations, we generated XGB-Mor that predicts the target molecular properties for the Morgan (Mor) fingerprints of the atom-level molecular structures.

Table 2: Characteristics of eight benchmark molecular datasets that contain the atom-level molecular structures and their experimentally observed target properties.

| Application Category | Dataset | Target Property | # of Molecules |
|---|---|---|---|
| Physicochemistry | Lipop [31] | Lipophilicity | 4,200 |
| | ESOL [56] | Aqueous solubility | 1,128 |
| | ADMET [57] | Aqueous solubility | 4,801 |
| Toxicity | IGC50 [58] | Tetrahymenapyriformis toxicity | 1,791 |
| | LC50 [58] | Fathead minnow toxicity | 822 |
| | LD50 [58] | Oral rat toxicity | 7,412 |
| Pharmacokinetics | LMC-H [31] | Liver microsomal clearance in human | 5,347 |
| | LMC-R [31] | Liver microsomal clearance in rat | 2,165 |

- **XGB-FC**: We generated XGB-FC by combining XGB with the functional-class (FC) [42] fingerprints of the input molecules. The FC fingerprint represents the atom-level molecular structures based on their functional substructures and atoms.

- **XGB-MK**: We also generated XGB-MK based on the MACCS Key (MK) fingerprint, which is one of the most commonly used molecular representations. MACCS key encodes the atom-level molecular structures based on 166-bits binary patterns.

- **GIN** [36]: Graph isomorphism network (GIN) is an effective framework for graph representation learning based on graph isomorphism test.

- **EGC** [37]: Efficient graph convolution (EGC) is an isotropic GNN based on adaptive filters and aggregation fusion in the node aggregation phase. EGC outperformed common anisotropic GNNs, such as graph attention networks, on benchmark datasets.

- **MPNN** [38]: Message passing neural network is a unified framework of node and edge convolution methods for learning molecular representations on quantum chemistry.

- **CGCNN** [39]: Crystal graph convolutional neural network (CGCNN) was proposed to predict the physical properties of the atomic structures of chemical compounds. It achieved state-of-the-art prediction accuracies in various chemical applications.

- **SchNet** [3]: It is a convolutional neural network for learning molecular representations based on quantum interactions in molecules. It has been widely used as a baseline model in various chemical applications [3, 59].

- **MEGNet** [40]: MatErials Graph Network (MEGNet) was proposed to predict the physical and chemical properties of molecular and crystal structures. It significantly improved the prediction accuracy by propagating the global state of the atomic structures through a message passing process between the atoms.

- **UniMP** [41]: Unified message passaging (UniMP) is a transformer-based GNN. UniMP showed state-of-the-art prediction capabilities by incorporating feature and label propagation at both training and inference time based on the transformer architecture.

- **DimeNet++** [33]: DimeNet++ is a variant of the original DimeNet [60]. DimeNet++ was more 10% more accurate than DimeNet on the QM9 dataset by solving information bottleneck in hierarchical representation leanring on molecular grapsh.

- **PhysChem** [34]: PhysChem was proposed to predict molecular properties by integrating the latent molecular embeddings from physical and chemical perspectives.

- **M3GNet** [35]: M3GNet is a graph neural network to learn inter-atomic potentials in molecular structures. It showed state-of-the-art prediction accuracy on several tasks in chemical applications by learning three-body atomic interactions of the molecular.

## Appendix 4: Implementation Details of HEDMoL

We converted a molecular structure into an attributed graph $G = (\mathcal{V}, \mathcal{U}, \mathbf{X}, \mathbf{E})$, where $\mathcal{V}$ is the set of nodes (i.e., atoms), $\mathcal{U}$ is the set of edges (i.e., chemical bonds), $\mathbf{X} \in \mathbb{R}^{|\mathcal{V}| \times d}$ is a $d$-dimensional

Table 3: Hyperparameter settings of HEDMoL for each benchmark molecular dataset. $l_a$ and $l_e$ mean the dimensionality of the graph embeddings for the atom-level molecular graph $G_a$ and the electron-derived substructure graph $G_e$.

| Dataset | Batch Size | Weight Regularization Coefficient | $\psi_a$ | $\psi_e$ | $l_a$ | $l_e$ | $\alpha$ | $\lambda$ |
|---------|-----------|-----------------------------------|----------|----------|-------|-------|----------|-----------|
| Lipop | 64 | 5e-6 | EGC | GIN | 32 | 32 | 0.2 | 1.0 |
| ESOL | 64 | 5e-6 | EGC | GIN | 16 | 16 | 0.2 | 1.0 |
| ADMET | 64 | 5e-6 | EGC | GIN | 16 | 16 | 0.2 | 1.0 |
| IGC50 | 32 | 5e-6 | EGC | GIN | 16 | 16 | 0.2 | 0.4 |
| LC50 | 32 | 5e-6 | SchNet | GIN | 16 | 16 | 0.1 | 0.2 |
| LD50 | 64 | 1e-6 | SchNet | GIN | 16 | 16 | 0.2 | 1.0 |
| LMC-H | 128 | 5e-6 | SchNet | GIN | 16 | 16 | 0.2 | 1.0 |
| LMC-R | 32 | 5e-6 | SchNet | GIN | 32 | 32 | 0.2 | 1.0 |

node-feature matrix, and $\mathbf{E} \in \mathbb{R}^{|\mathcal{U}| \times l}$ is an $l$-dimensional edge-feature matrix. We used the pre-defined 200-dimensional atomic embeddings [61] with atomic features determined by local molecular environments to define the input node-feature matrix $\mathbf{X}$. To assign the edge features, we followed the popular implementation that defines the edge features between the atoms as the 22-dimensional one-hot encoding of the bonding types [2, 40]. The pre-defined bonding types were provided in RDKit[2], which is a popular cheminformatics library in computational chemistry. We used the junction tree algorithm provided by PyTorch Geometric[3] for the implementation of the substructure decomposition step in HEDMoL.

We used the grid search to optimize the training hyperparameters of the competitor GNNs and HEDMoL, such as batch size and weight regularization coefficient. The hyperparameter settings of HEDMoL for each dataset are presented in Table 3. The initial learning rate in the training of HEDMoL was fixed to 5e-4 for all datasets. In the implementation of HEDMoL, we used GIN as the GNN-based embedding network for the electron-derived substructure graphs for all datasets. However, we used EGC or SchNet as the GNN-based embedding network for the atom-level molecular graph, as shown in Table 3. All competitor GNNs were constructed by stacking one dense layer for node-feature embedding, two node aggregation layers with layer normalization [62], and two dense layers for prediction. The number of hidden channels was fixed to 256 for all competitor GNNs. HEDMoL used GIN, EGC, and SchNet with the same architecture of the competitor GNNs, but the number of hidden channels between the node aggregation and dense layers was fixed to 128 for fair comparisons on a similar number of model parameters. HEDMoL was implemented with Python 3.9. All implementations and experiment scripts are written by PyTorch 2.0.0+cu117 and PyTorch Geometric 2.3.1 with CUDA 11.7. All experiments were conducted in a machine with Intel i9-12900K CPU, 128G memory, and NVIDIA GeForce RTX 3090 Ti.

For the knowledge extension of HEDMoL, we used the QM9 dataset as an external database. The QM9 dataset is a well-known calculation dataset containing pre-calculated electron-level information about small organic molecules. In the implementation of HEDMoL, we select molecules that consist of six or fewer atoms from the QM9 dataset to obtain a subset containing 685 molecules. This is because we aim to perform knowledge extension based on small molecules since the electron-level calculation error is small for small molecules. In the knowledge extension, the transferred substructure feature $\mathbf{X}_{e,k}$ is defined as a substructure-feature matrix $\mathbf{X}_e$ is defined as a 12-dimensional vector containing physically calculated electronic attributes: dipole moment, isotropic polarizability, HOMO, MUMO, HOMO-LUMO gap, electronic spatial extent, vibrational energy, internal energy at 0 K and 298.15 K, enthalpy, free energy, and heat capacity.

## Appendix 5: Prediction Accuracy of HEDMoL Architectures with Junction Tree and BRICS Decomposition Algorithms

There are several choices in the implementation of the substructure decomposition in HEDMoL. In the implementation of HEDMoL, we used the junction tree algorithm to decompose the input

---

[2]https://www.rdkit.org
[3]https://pytorch-geometric.readthedocs.io

Table 4: Measured R2-scores of HEDMoL and HEDMoL-BC.

| Method | Lipop | ESOL | ADMET | IGC50 | LC50 | LD50 | LMC-H | LMC-R |
|--------|-------|------|-------|-------|------|------|-------|-------|
| HEDMoL-BC | 0.731 (0.054) | 0.915 (0.021) | 0.862 (0.007) | 0.836 (0.016) | 0.620 (0.081) | N/A | 0.532 (0.013) | 0.622 (0.039) |
| HEDMoL | **0.759** **(0.043)** | 0.914 (0.016) | 0.865 (0.014) | 0.840 (0.010) | **0.663** **(0.053)** | **0.572** **(0.035)** | 0.551 (0.008) | 0.639 (0.035) |

molecular graphs into the subgraphs. However, there is an alternative implementation of HEDMoL based on the BRICS decomposition [21], which is the most common graph decomposition method in chemical science to split the molecular graph into the chemically-valid substructures. In this experiment, we conducted an experimental evaluation with a variant of HEDMoL, i.e., HEDMoL-BC, in which the junction tree algorithm in the substructure decomposition process is replaced with the BRICS decomposition [21].

As shown in Table 4, HEDMoL showed marginally better prediction accuracy than HEDMoL-BC on the most benchmark datasets. In particular, HEDMoL outperformed HEDMoL-BC on the Lipop and LC50 dataset. This experimental result comes from two reasons: (1) The BRICS decomposition cannot split the large molecules into sufficiently small substructures because it should guarantee the chemical validity of the decomposed substructures. (2) Many hydrogens are virtually added to the decomposed substructures for their chemical validity, and it can distort the structural information of the original molecules. Furthermore, we were not able to execute HEDMoL-BC on the LD50 dataset because the execution time of the BRICS decomposition exploded on several large molecules of the LD50 dataset. This experimental result implies the effectiveness of processing the molecular structures as an attributed graph rather than a chemical system.

Table 5: Test $R^2$-scores of HEDMoL for different implementations of the knowledge extension. FEATURE-G, FGSD, Graph2Vec, and GeoScattering are unsupervised graph embedding methods. We implemented the knowledge extension step of HEDMoL based on GeoScattering.

| Dataset | Morgan Fingerprint | FEATHER-G | FGSD | Graph2Vec | GeoScattering |
|---------|--------------------|-----------|------|-----------|---------------|
| Lipop | 0.734 (0.044) | 0.734 (0.042) | 0.741 (0.038) | 0.734 (0.047) | 0.736 (0.030) |
| ESOL | 0.909 (0.018) | 0.912 (0.019) | 0.914 (0.014) | 0.910 (0.016) | 0.915 (0.016) |
| ADMET | 0.865 (0.011) | 0.870 (0.010) | 0.864 (0.013) | 0.864 (0.010) | 0.861 (0.010) |
| IGC50 | 0.796 (0.018) | 0.835 (0.011) | 0.837 (0.021) | 0.831 (0.023) | 0.835 (0.017) |

## Appendix 6: Prediction Capabilities for Different Graph Embedding Methods in the Knowledge Extension

There are several choices of $\pi(S_k, G_i^s)$ in the implementation of the knowledge extension in HEDMoL. In this experiment, we evaluated the prediction capabilities of HEDMoL for different graph embedding methods in the knowledge extension step. We implemented HEDMoL with four different graph embedding methods: Morgan fingerprint[42], FEATHER-G [63], FGSD [64], Graph2Vec [65], and GeoScattering [23]. Note that we implemented $\pi(S_k, G_i^s)$ with the Morgan fingerprint based on the Tanimoto distance [66] for the Morgan fingerprints [42] of the molecules, which is a common approach for finding similar molecules in cheminformatics. The experimental evaluation was conducted on the Lipop, ESOL, ADMET, and IGC50 datasets, where the competitor GNNs and HEDMoL achieved R2 scores greater than 0.7. As shown in Table 5, HEDMoL with the graph embedding methods showed similar $R^2$-scores for all benchmark datasets because HEDMoL already assumes the matching errors in the knowledge extension, which was formalized by the Gaussian error $\epsilon_k$. As a result, HEDMoL was robust to the implementation choices of the knowledge extension.

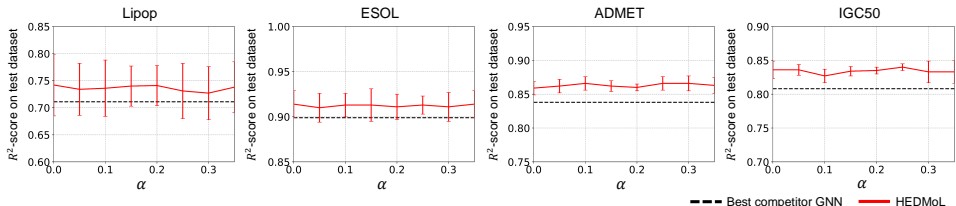

Figure 2: $R^2$-scores of HEDMoL on the Lipop, ESOL, ADMET, and IGC50 datasets for different values of $\alpha$.

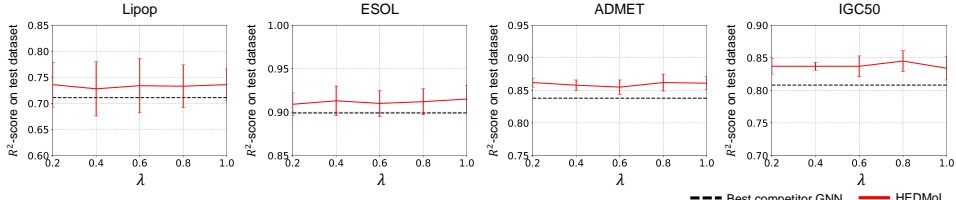

Figure 3: $R^2$-scores of HEDMoL on the Lipop, ESOL, ADMET, and IGC50 datasets for different values of $\lambda$.

## Appendix 7: Hyperparameter Analysis for $\alpha$ and $\lambda$

In the training process of HEDMoL, there are two hyperparameters $\alpha$ and $\lambda$ to control the structural approximation errors and the effect of the physical consistency regularization, respectively. In this experiment, we measured the $R^2$-scores of HEDMoL for different values of $\alpha$ and $\lambda$ on the Lipop, ESOL, ADMET, and IGC50 datasets. We selected the hyperparameter values of $\alpha$ and $\lambda$ in {0, 0.05, 0.1, 0.15, 0.2, 0.25, 0.3, 0.35} and {0.2, 0.4, 0.6, 0.8, 1.0}, respectively. Figs. 2 and 3 show the measured $R^2$-scores for different hyperparameter values. In this experiment, HEDMoL consistently achieved better $R^2$-scores than those of the best competitor GNNs for all hyperparameter values and benchmark datasets. In particular, HEDMoL outperformed the best competitor GNNs over the standard deviations on the ESOL, ADMET, and IGC50 datasets for all hyperparameter values. These experimental results show the robustness of HEDMoL to the hyperparameter values.

## Appendix 8: Representation Capabilities for External Databases of Different Molecular Scales

HEDMoL is a representation learning method based on knowledge transfer between the information on different scales. In this experiment, we evaluated the prediction capabilities of HEDMoL for the external calculation databases containing molecules of different sizes. We generated external calculation databases of different molecular scales by selecting the molecules from the QM9 dataset based on the number of atoms. We generated six external calculation databases that contain molecules consisting of two to $C$ atoms, where $C = \{3, 4, 5, 6, 7, 8\}$. Note that the knowledge extension of HEDMoL was implemented with $C = 6$. Table 6 shows the measured $R^2$-scores of HEDMoL on the external calculation databases of different molecular sizes.

Table 6: $R^2$-scores of HEDMoL for the external datasets of different molecular scales. $C$ indicates the maximum number of atoms of the molecules in the subset of the QM9 dataset. HEDMoL was implemented with the QM9 subset of $C = 6$.

| Dataset | $C = 3$ | $C = 4$ | $C = 5$ | $C = 6$ | $C = 7$ | $C = 8$ |
|---------|---------|---------|---------|---------|---------|---------|
| Lipop | 0.732 (0.037) | 0.723 (0.053) | 0.735 (0.047) | 0.736 (0.030) | 0.738 (0.040) | 0.736 (0.037) |
| ESOL | 0.914 (0.018) | 0.911 (0.016) | 0.915 (0.015) | 0.915 (0.016) | 0.916 (0.015) | 0.914 (0.019) |
| ADMET | 0.867 (0.007) | 0.862 (0.015) | 0.868 (0.012) | 0.861 (0.010) | 0.867 (0.012) | 0.865 (0.014) |
| IGC50 | 0.829 (0.017) | 0.833 (0.012) | 0.828 (0.016) | 0.835 (0.017) | 0.825 (0.018) | 0.831 (0.016) |

As shown in the table, the $R^2$-scores of HEDMoL were robust to the changes in the molecular scales of the external datasets because the junction tree algorithm in the knowledge extension of HEDMoL decomposes the input molecules into tiny substructures. For this reason, most decomposed substructures would have been sufficiently approximated by the small molecules in the QM9 subsets of $C = \{3, 4, 5\}$. Similarly, the changes on the $R^2$-scores were marginal for $C = \{7, 8\}$ because most decomposed substructures will be replaced with the small molecules containing 3-5 atoms even on the QM9 subsets containing larger molecules. This experimental result implies that preparing an external dataset containing enough small molecules is not a challenging task because possible molecular structures decrease exponentially as the number of atoms in the molecules decreases.

## Appendix 9: Execution Time of the Training and Inference Processes

HEDMoL inevitably requires additional computation costs in the training and inference phases to execute the hierarchical representation learning with two embedding networks $\psi_a$ and $\psi_e$. Moreover, calculating the regularization terms of the physical consistency also causes additional computations. We compared the execution time of HEDMoL with EGC, GIN, MPNN, and SchNet on the LC50, Lipop, and LD50 datasets. MPNN and SchNet showed the best prediction accuracy on the benchmark datasets. EGC, GIN, and SchNet were used as the embedding networks of HEDMoL. The LC50, Lipop, and LD50 datasets contain 822, 4,200, and 7,412 molecules, respectively. This experiment to measure the execution time was conducted in a machine with Intel i9-12900K CPU, 128G memory, and NVIDIA GeForce RTX 3090 Ti.

Table 7: Execution time per epoch (sec/epoch) on the LC50, Lipop, and LD50 datasets. The LC50, Lipop, and LD50 datasets contain 822, 4,200, 7,412 molecules, respectively.

| Dataset | Training | | | | | Inference | | | | |
|---|---|---|---|---|---|---|---|---|---|---|
| | EGC | GIN | MPNN | SchNet | HEDMoL | EGC | GIN | MPNN | SchNet | HEDMoL |
| LC50 | 0.087 | 0.077 | 0.091 | 0.172 | 0.229 | 0.014 | 0.013 | 0.014 | 0.022 | 0.035 |
| Lipop | 0.643 | 0.558 | 0.619 | 1.650 | 1.371 | 0.099 | 0.090 | 0.099 | 0.191 | 0.228 |
| LD50 | 0.929 | 0.828 | 0.902 | 2.038 | 2.468 | 0.136 | 0.125 | 0.133 | 0.237 | 0.373 |

Table 7 shows the execution time of MPNN, SchNet, and HEDMoL in the training and inference processes. Obviously, HEDMoL requires more execution time in the training and inference processes because it has two GNN-based embedding networks. However, the execution time of HEDMoL linearly increased as the sum of two embedding networks. For example, the execution time of HEDMoL on the LC50 dataset is similar to the sum of the execution times of GIN and SchNet, as shown in $0.229 \approx 0.077 + 0.172$ of the training process on the LC dataset. Also, the execution time of HEDMoL on the Lipop dataset is similar to the sum of the execution times of EGC and GIN, as shown in $1.371 \approx 0.643 + 0.558$ of the training process on the Lipop dataset. For the inference time, the execution time of HEDMoL linearly increased.

## Appendix 10: Evaluation Results on Mean Absolute Error

Table 8 presents the measured mean absolute errors (MAEs) of the competitor methods and HEDMoL on the benchmark molecular datasets. As with the result measured by $R^2$-score, HEDMoL showed the best performance for all benchmark datasets, i.e., HEDMoL achieved the lowest MAEs for all benchmark datasets.

## Appendix 11: Prediction Accuracy for Different Types of Electron-Level Information

The representation learning results of HEDMoL are affected by the electron-level features provided in the external database. For more investigation, we measured the prediction accuracy of HEDMoL for different types of electron-level information in the QM9 dataset. In this experiment, we categorized the electron-level features in the QM9 dataset into three categories: energy-related features, temperature-dependent features, and features in other categories. There are 7, 2, and 3 electron-level features

Table 8: Measured MAEs of the competitor methods and HEDMoL on the benchmark molecular datasets. Input type means the required data format of the input molecules for each method. The highest MAE for each benchmark dataset has been remarked in bold, and the standard deviation of the MAEs is presented in parentheses.

| Input Type | Method | Lipop | ESOL | ADMET | IGC50 | LC50 | LD50 | LMC-H | LMC-R |
|---|---|---|---|---|---|---|---|---|---|
| Molecular Fingerprint | XGB-Mor [42] | 0.608 (0.012) | 0.908 (0.039) | 0.855 (0.018) | 0.648 (0.017) | 0.876 (0.061) | 0.475 (0.008) | 0.358 (0.006) | 0.368 (0.027) |
| | XGB-FC [42] | 0.586 (0.010) | 0.874 (0.064) | 0.855 (0.017) | 0.468 (0.021) | 0.776 (0.020) | 0.484 (0.006) | 0.357 (0.005) | 0.365 (0.007) |
| | XGB-MK [43] | 0.610 (0.014) | 0.740 (0.047) | 0.773 (0.017) | 0.444 (0.012) | 0.770 (0.139) | 0.491 (0.015) | 0.364 (0.004) | 0.373 (0.018) |
| 3D molecular Graph | DimeNet++ [33] | N/R | 0.541 (0.040) | N/R | 0.385 (0.023) | 0.681 (0.024) | N/A | 0.455 (0.021) | N/R |
| | PhysChem [34] | 0.453 (0.019) | 0.563 (0.038) | N/A | 0.318 (0.021) | 0.675 (0.038) | N/A | N/A | N/R |
| | M3GNet [35] | N/A | 0.531 (0.035) | N/A | 0.451 (0.021) | 0.681 (0.045) | N/A | N/A | 0.378 (0.011) |
| 2D Molecular Graph | GIN [36] | 0.437 (0.024) | 0.443 (0.040) | 0.575 (0.017) | 0.311 (0.012) | 0.663 (0.038) | 0.475 (0.016) | 0.365 (0.007) | 0.381 (0.008) |
| | EGC [37] | 0.436 (0.016) | 0.434 (0.016) | 0.590 (0.012) | 0.306 (0.022) | 0.644 (0.039) | 0.486 (0.005) | 0.367 (0.005) | 0.378 (0.009) |
| | MPNN [38] | 0.438 (0.012) | 0.453 (0.017) | 0.611 (0.011) | 0.312 (0.011) | 0.670 (0.053) | 0.500 (0.014) | 0.363 (0.011) | 0.382 (0.014) |
| | CGCNN [39] | 0.442 (0.020) | 0.441 (0.041) | 0.599 (0.012) | 0.301 (0.019) | 0.686 (0.046) | 0.490 (0.015) | 0.368 (0.005) | 0.368 (0.006) |
| | SchNet [3] | 0.560 (0.013) | 0.554 (0.042) | 0.606 (0.020) | 0.390 (0.022) | 0.677 (0.019) | 0.483 (0.010) | 0.359 (0.007) | 0.410 (0.016) |
| | MEGNet [40] | 0.537 (0.019) | 0.474 (0.024) | 0.601 (0.025) | 0.355 (0.018) | 0.662 (0.054) | 0.478 (0.011) | 0.371 (0.008) | 0.366 (0.010) |
| | UniMP [41] | 0.449 (0.020) | 0.472 (0.030) | 0.599 (0.021) | 0.320 (0.024) | 0.682 (0.035) | 0.497 (0.009) | 0.373 (0.011) | 0.376 (0.016) |
| | **HEDMoL** | **0.427 (0.037)** | **0.427 (0.027)** | **0.554 (0.026)** | **0.287 (0.007)** | **0.603 (0.036)** | **0.451 (0.013)** | **0.335 (0.004)** | **0.344 (0.016)** |

Table 9: R2-scores of HEDMoL for different types of electron-level features in the QM9 dataset.

| Dataset | Lipop | ESOL | ADMET | IGC50 | LC50 | LD50 | LMC-H | LMC-R |
|---|---|---|---|---|---|---|---|---|
| Energy-related features | 0.736 (0.049) | 0.918 (0.019) | 0.866 (0.015) | 0.832 (0.011) | 0.628 (0.069) | 0.562 (0.027) | 0.541 (0.009) | 0.619 (0.032) |
| Temperature-dependent features | 0.730 (0.053) | 0.915 (0.020) | 0.863 (0.012) | 0.837 (0.012) | 0.656 (0.045) | 0.582 (0.026) | 0.537 (0.011) | 0.624 (0.037) |
| Features in other categories | 0.720 (0.051) | 0.919 (0.016) | 0.868 (0.013) | 0.836 (0.008) | 0.642 (0.076) | 0.557 (0.029) | 0.533 (0.009) | 0.621 (0.036) |
| All features (HEDMoL) | 0.759 (0.043) | 0.914 (0.016) | 0.865 (0.014) | 0.840 (0.010) | 0.663 (0.053) | 0.572 (0.035) | 0.551 (0.008) | 0.639 (0.035) |

in the energy-related, temperature-dependent, and other categories, respectively. Table 9 shows the R2-scores of HEDMoL using only the electron-level features in each category. "All features" in the table means HEDMoL using all electron-level features in the QM9 dataset, which is the same model as HEDMoL in the paper.

As shown in Table 9, the prediction model using all electron-level features showed better R2-scores on the Lipop, IGC50, LC50, LD50, LMC-H, and LMC-R datasets, because the molecular properties of toxicity and pharmacokinetics are comprehensively related to the electron-level features of the molecules. However, we were not able to observe the performance improvement on the ESOL and ADMET datasets, which contain the aqueous solubilities of the molecules. This result is consistent with existing experimental results that the aqueous solubilities are usually dependent on the molecular scale rather than electronic distributions and energies [67].

# Appendix 12: Prediction Accuracy for Input Molecules of Different Molecular Scales

The LMC-H and LMC-R datasets contain relatively large molecules where the average number of atoms and edges are greater than 50 and 110, respectively. In these datasets, the prediction accuracy of the existing GNNs was significantly degraded, and the simple XGB-based methods outperformed them, as shown in Table 1 of the manuscript. Our results align with existing study reporting the performance degradations of GNNs on large molecular graphs [48]. On the other hand, HEDMoL showed an improved prediction accuracy and outperformed all competitor GNNs and XGB-based methods on the LMC-H and LMC-R datasets, which contain relatively large molecules. This result comes from the substructure decomposition-based graph representation learning of HEDMoL in Sections 3.3-3.5. It is consistent with existing experimental results that the subgraph decomposition approach can improve the representation and generalization capabilities of the graph embedding models [6, 7].

Table 10: Prediction accuracy and improvement of HEDMoL according to the molecular sizes. The average molecular size is presented in the parenthesis of each dataset name. The average number of atoms was presented together with the dataset name.

| Method | IGC50 (19.29) | LC50 (22.32) | ESOL (25.63) | ADMET (26.76) | LD50 (31.30) | Lipop (48.51) | LMC-R (53.13) | LMC-H (54.54) |
|---|---|---|---|---|---|---|---|---|
| R2-score of the best competitor (A) | 0.808 | 0.587 | 0.899 | 0.838 | 0.526 | 0.703 | 0.617 | 0.505 |
| R2-score of HEDMoL (B) | 0.840 | 0.663 | 0.914 | 0.865 | 0.572 | 0.759 | 0.639 | 0.551 |
| Accuracy improvement ($100 \times ((B - A)/A)$) | 3.960 | 12.947 | 1.669 | 3.222 | 8.745 | 7.966 | 3.566 | 9.109 |

Furthermore, we quantitatively measured the accuracy improvement of HEDMoL according to the molecular size in molecular property prediction, as shown in Table 10. We sorted the benchmark molecular datasets according to the average number of atoms in the molecules. As shown in Table 10, the accuracy improvement of HEDMoL does not decreased in the LMC-R and LMC-H datasets, which contain relatively large molecules. The accuracy improvement of HEDMoL was independent to the molecular size, and their Pearson correlation coefficient was 0.103. We will add descriptions of the statistics of the molecules in each benchmark dataset. Also, we will supplement the discussion section about the accuracy improvement of HEDMoL and the molecular size.

# Appendix 13: Ablation Study on HEDMoL

In this experiment, we conducted an ablation study of HEDMoL to investigate the effects of each component on the prediction capabilities of HEDMoL. We compared the $R^2$-scores of EGC and GIN because HEDMoL is equivalent to the embedding networks $\psi_a$ or $\psi_e$ if all the proposed components are removed from HEDMoL. Table 11 shows the experimental results on HEDMoL. To implement KE without HRL, we used $\mathbf{z} = \mathbf{z}_a \oplus \mathbf{z}_e$ as the final molecular embedding instead of $\mathbf{z} = \mathbf{z}_a \oplus \mathbf{z}_c$. We observe that the $R^2$-scores of the prediction models notably increased after implementing KE for all datasets, as shown in Table 11. This result explicitly shows the effectiveness of our knowledge

Table 11: Ablation studies on HEDMoL. Each abbreviation means the components of HEDMoL as follows. KE: knowledge extension. HRL: hierarchical molecular representation learning. PCR: energy-based physical consistency regularization. In this abliation study, KE+HRL+PCR denotes HEDMoL.

| Dataset | EGC | GIN | KE | KE+HRL | KE+PCR | KE+HRL+PCR |
|---|---|---|---|---|---|---|
| Lipop | 0.708 (0.043) | 0.702 (0.031) | 0.722 (0.051) | 0.726 (0.049) | 0.731 (0.044) | 0.759 (0.043) |
| ESOL | 0.896 (0.017) | 0.897 (0.022) | 0.913 (0.015) | 0.913 (0.015) | 0.915 (0.017) | 0.914 (0.016) |
| ADMET | 0.838 (0.012) | 0.833 (0.017) | 0.862 (0.014) | 0.864 (0.015) | 0.860 (0.013) | 0.865 (0.014) |
| IGC50 | 0.808 (0.029) | 0.799 (0.021) | 0.827 (0.018) | 0.833 (0.017) | 0.826 (0.014) | 0.840 (0.010) |
| LC50 | 0.575 (0.045) | 0.543 (0.080) | 0.631 (0.091) | 0.640 (0.084) | 0.603 (0.061) | 0.663 (0.053) |

extension approach for propagating the electron-level attributes to the atom-level molecular structures. Although the improvements by KE+HRL and KE+PCR were not observed in the experiment, the $R^2$-scores were improved, and their standard deviations decreased by KE+HRL+PCR on the Lipop, IGC50, and LC50 datasets.

## Appendix 14: Prediction Capabilities on Small Training Datasets

Since conducting chemical experiments to obtain the experimentally labeled data is time-consuming and labor-intensive, the lack of training data is one of the main challenges of machine learning in chemical applications [15, 16]. As described in the knowledge extension step of Section 2.4, HEDMoL inherently has the ability to extend the electron-level knowledge regarding small molecules to unseen large molecules, which is beneficial in constructing an accurate prediction model on small training datasets. In this experiment, we compared $R^2$-scores of the competitor methods and HEDMoL over different sizes of training datasets to demonstrate the prediction capabilities of HEDMoL on small training datasets.

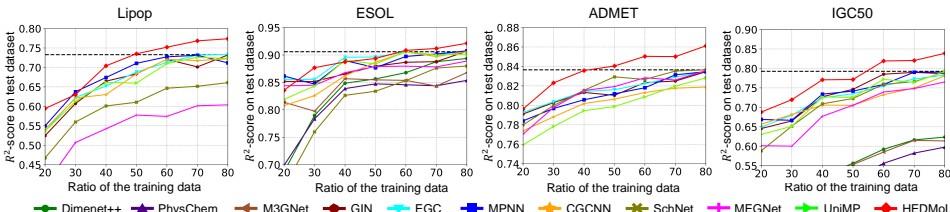

Figure 4: $R^2$-scores for different sizes of training data. The black dotted line indicates the $R^2$-score of the best competitor method on the 80% training data.

Fig. 4 shows the $R^2$-scores of the competitor methods and HEDMoL for different sizes of training datasets. We measured the $R^2$-scores on the Lipop, ESOL, ADMET, and IGC50 datasets in which HEDMoL and most competitor GNNs achieved the $R^2$-scores greater than 0.6. We did not measure the $R^2$-scores of the XGB- and 3D structure-based methods because most of them failed on small training datasets. Obviously, we were able to observe that the prediction accuracy tends to be improved as the size of the training dataset increases. However, HEDMoL showed higher $R^2$-scores than the competitor GNNs for all sizes of the training datasets on the Lipop, ADMET, and IGC50 datasets. Furthermore, HEDMoL already achieved comparable $R^2$-scores with each best competitor method on the 80% training data (black dotted line) on the 50%, 40%, and 60% training data of the Lipop, ADMET, and IGC0 datasets, respectively. These experimental results show the practical potential of HEDMoL in real-world chemical applications, which usually suffer from the lack of training data.

