# OpenReview forum: "Electron-Derived Molecular Representation Learning for Real-World Molecular Physics"
_NeurIPS.cc/2023/Workshop/AI4Science — NeurIPS2023-AI4Science Poster_

### Official Review · Reviewer_zpg4 · 2023-10-13
**Review of Electron-Derived Molecular Representation Learning for Real-World Molecular Physics**

**Rating:** 8
**Confidence:** 4

**Review:**

Summary:
This paper uses electron-derived molecular representations to achieve SOTA results on a wide variety of molecular benchmark datasets. It does so by QM9 as a library of pre-calculated electron-level information for small molecular fragments that they use to construct their electron-derived molecular representation.

Clarity:
This paper is well written and clearly motivated. Their architecture is communicated well and their results are clearly shown. Some of the equations could be clarified. Equation (12) and (13) should use regularisation against $E_{p,k} + \epsilon_k$ not $E_{a,k}$ and $E_{e,k}$ as it is currently written.

Quality:
They preform comprehensive experiments on a wide variety of molecular benchmarks to demonstrate the achievements of their model.

Strengths:
This is a novel approach to improving molecular representations, by using electron-level information for molecular motifs. This avoids the issue of high computation costs for computing electron-level information for large molecules, while leveraging electron-level information for smaller molecules. They also demonstrate SOTA performance on a variety of benchmarks.

Weaknesses:
Some further clarification in the notation would be helpful to guide readers.

---

### Official Review · Reviewer_SUm4 · 2023-10-21
**Review of 'Electron-Derived Molecular Representation Learning for Real-World Molecular Physics'**

**Rating:** 8
**Confidence:** 4

**Review:**

In this paper, the authors propose an interesting approach of informing electronic information via substructure decomposition and similarity search in QM9. Overall, an improved model performance is found when evaluating this approach on several experimental datasets.

Pros: a thorough description is presented on the framework and technical details, and the proposed approach is interesting and giving good model performance on a comprehensive list of datasets.

A few more comments: (1) only R2 statistics are reported for the benchmarking results, can authors also include RMSE? (2) I would like to see an ablation study on w/o electronic information, while keeping the other part of network the same. (3) can authors show some examples of knowledge extension step? I am most interested in the molecule pair with low similarity score, would that be a potential limitation of the proposed work here? Would all substructures be able to successfully recover with their similar structures in QM9?